# Laundry Fabric Classification in Vertical Axis Washing Machines Using Data-Driven Soft Sensors

**Marco Maggipinto [1,\*], Elena Pesavento [2], Fabio Altinier [2], Giuliano Zambonin [1,2], Alessandro Beghi [1,3] and Gian Antonio Susto [1,3]** 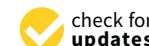

[1]   Department of Information Engineering, University of Padova, 35131 Padova, Italy; giuliano.zambonin@electrolux.com (G.Z.); beghi@dei.unipd.it (A.B.); sustogia@dei.unipd.it (G.A.S.)
[2]   Electrolux Italia S.p.a., PN 33080 Porcia, Italy; elena.pesavento@electrolux.com (E.P.); fabio.altinier@electrolux.it (F.A.)
[3]   Human Inspired Technology Research Centre, University of Padova, 35121 Padova, Italy
\*   Correspondence: marco.maggipinto@studenti.unipd.it; Tel.: +39-049-827-7760

**Abstract:** Embedding household appliances with smart capabilities is becoming common practice among major fabric-care producers that seek competitiveness on the market by providing more efficient and easy-to-use products. In Vertical Axis Washing Machines (VA-WM), knowing the laundry composition is fundamental to setting the washing cycle properly with positive impact both on energy/water consumption and on washing performance. An indication of the load typology composition (cotton, silk, etc.) is typically provided by the user through a physical selector that, unfortunately, is often placed by the user on the most general setting due to the discomfort of manually changing configurations. An automated mechanism to determine such key information would thus provide increased user experience, better washing performance, and reduced consumption; for this reason, we present here a data-driven soft sensor that exploits physical measurements already available on board a commercial VA-WM to provide an estimate of the load typology through a machine-learning-based statistical model of the process. The proposed method is able to work in a resource-constrained environment such as the firmware of a VA-WM.

**Keywords:** household appliances; machine learning; regularization; soft sensors; sustainability; vertical axis washing machines

## 1. Introduction

Household appliances became extremely popular during the last century thanks to mass production and the consequent affordable prices [1]. Efficiency and usability of such appliances have dramatically improved during the years; however, there are still many challenges for manufacturers, especially in the context of sustainability and user experience of such products. In recent years, embedding smart capabilities (e.g., speech recognition and automatic decision making) into household devices is becoming more and more popular [2] and essential to be competitive on the market, a trend that has the potential to revolutionize the way we use and interact with such products. On the other hand, strict environmental laws push manufacturers to develop innovative solutions to limit the impact of their product on the environment. Both in the European and American markets, it is mandatory to apply an Energy Label/Energy Star sticker on every product that indicates its energy efficiency so that customers can make an informed choice of the products their are buying. Of course, Energy Label/Energy Star influences consumer choice in making a purchase [3], making it extremely important for the manufacturers to get high scores on such rankings.

In fabric-care appliances—washing machines (WM), tumble dryers (TD), and washer dryers (WD)—the impact on the environment is mainly determined by energy and water consumption; hence,

manufacturers put considerable effort into optimizing washing/drying cycles. In this regards, the type of laundry (e.g., cotton, synthetic, etc.) that the user loads in WMs and WDs plays a fundamental role in determining the optimal washing cycle configuration due to the different water absorption properties and resistance of the fabric that, in a worst-case scenario, may be even damaged when washed incorrectly (e.g., high temperature). We remark that the main source of variability in drying and washing processes is represented by the laundry inserted in the appliance. Most of the WMs and WDs available on the market require this information to be provided by the user through a physical selector that is often left on the most general settings in order to avoid this manual operation with undesirable consequences on the washing performance. For this reason, automating the process of load detection would have considerable impact on the product efficiency and usability, making it more appealing on the market. However, load typology is a quality that cannot be measured by a physical sensor; it is thus necessary to exploit indirect measures that provide information on physical properties of the laundry related to its typology (e.g., water absorption). A Soft Sensor (SS) [4] is a technology that allows for the estimation of the value of a quantity that is too costly or even impossible to measure from indirect sensor measurements, making it well suited for the typology detection task at hand. In literature, SSs are typically divided into two categories [5]:

- Model driven—SSs that exploit the physical model of the process to perform the prediction.
- Data driven—SSs that build a statistical model of the process by leveraging historical data. Here Machine Learning (ML) methodologies are often exploited.

In this paper, due to the complexity of the physical process at hand, we propose a data-driven SS for laundry typology detection. As a reference-use case, we have developed such SS for a Vertical Axis WM (VA-WM), the typical WM of the American market. The SS exploits measurements already available on-board a commercial VA-WM, in particular, speed and torque from the electric engine, without the need to equip the machine with other physical sensors that would be unfeasible in terms of production costs. The proposed SS is based on supervised-learning [6] methods where a set of input data (sensors measurements) with the associated correct output (laundry typology) is available; a predictive function is then fit to such data by minimizing the opportune loss function on a training set. In our work, a set of laboratory tests have been performed with known load in order to collect the required data.

Due to the limited resources available in the VA-WM firmware, our approach relies on simple regularization techniques. We also propose a more advanced solution based on hierarchical classification methods that, even if computationally unfeasible for our application, can be of interest with future hardware configurations.

Data-driven SS are common in industrial environments such as semiconductor manufacturing [7,8], chemical [9–11], and automotive [12,13]. The methodologies employed in the literature usually vary from simple regression/classification techniques such as linear regression [14] and Bayesian Networks [11] to more complex neural-network-based algorithms [7]. However, in fabric-care home appliances, the application of SSs is limited to load quantity [15–18] and humidity estimation [19,20] and the resource constrained environment makes the problem challenging and the possible solution limited to simple ones.

The most discriminative property of different laundry typologies is the quantity of water absorbed during the washing cycle; however, this is affected also by the quantity of clothes inserted in the WM. For this reason, in Reference [18], we proposed a SS for load-weight estimation that provided accurate predictions exploiting on-board sensor measurements. In this work, we assume the weight to either be known or estimated through another load-weight SS.

The reminder of this paper is organized as follows: In Section 2, we describe the methodologies employed in the development of our SS; in Section 3, we briefly introduce VA-WM and their washing cycle; in Section 4, we detail the proposed algorithm; and we show the results in Section 5. In Section 6, we draw the conclusions and discuss future works.

All the data employed in the development of this work have been provided by Electrolux Italia S.p.A. and cannot be shared for confidentiality reasons.

## 2. Methodologies

Supervised-learning techniques have been extensively studied in the past thanks to their straightforward applicability to many prediction problems where a "labeled" dataset is available [21]. Given a set of input data $x_i \in \mathbb{R}^p$ $i \in \{1, \cdots, n\}$, where $p$ is the number of features or predictors and $n$ is the number of available observations, with the associated correct output label $y_i$ that can either be real $y_i \in \mathbb{R}$ or categorical $y_i \in \{0, \cdots, K-1\}$ (respectively if the output is continuous or belongs to a finite set of classes, as in the case of load typology), the goal of supervised learning is to find a function $y = f_\theta(x)$ parametrized by $\theta$ that approximates the real input–output relation of the data, providing an estimation model that can be used to estimate the output for previously unseen input values. Typically, in simple supervised methods, the data are organized in a design matrix $X \in \mathcal{R}^{n \times p}$ where each row corresponds to an observation; this allows to express optimization algorithms in a convenient vectorial form. The simplest classification algorithm for binary output values $y_i \in \{0, 1\}$ is called logistic regression and is at the foundation of the proposed soft sensor.

### 2.1. Logistic Regression and Regularization

Logistic regression models the input–output relation by means of the so-called logistic function $y = \frac{1}{1-e^{\theta^T x}}$ that assumes values in the open interval $(0, 1)$ and, hence, provides a probabilistic interpretation of the model that can be viewed as an estimate of the conditional probability $p(y = 1|x)$. Typically, the classification is performed by placing a threshold on the output value in order to assign every element to either class 0 or 1. Training the model requires finding the value for the parameters vector that minimizes the cross entropy loss [6] defined as follows:

$$\ell(\theta) = \sum_1^n y_i log(\hat{y}_i) + (1 - y_i) log(1 - \hat{y}_i) \tag{1}$$

Here, $y_i$ and $\hat{y}_i$ are respectively the real and predicted output. Minimization is usually achieved through iterative algorithms such as Gradient Descent [22].

The simple logistic regression algorithm as explained above suffers from high variance whenever the number of features is higher than the number of data available or substantial collinearity is present in the data. This phenomena is often referred to as overfitting and happens whenever the model fits very well the training set but fails at predicting accurately previously unseen data, making it useless for any real application. To mitigate this phenomenon, regularization techniques that encourage the model towards simpler but more general functions are employed in the optimization procedure. This is achieved by adding a regularization term $\mathcal{R}(\theta)$ that penalizes the norm of the parameters vector. The most common regularization techniques are as follows:

- Ridge [23], that penalizes the $L_2$ norm of the parameters vector $\mathcal{R}(\theta) = ||\theta||_2$;
- Least Absolute Shrinkage and Selection Operator (LASSO) [24], that penalized the $L_1$ norm of the parameters vector $\mathcal{R}(\theta) = ||\theta||_1$

Typically, the penalty is weighted by a hyperparamenter $\lambda$ that allows to tune the amount of regularization in order to find the best trade off between prediction error on the training set and prediction capabilities on unseen data. The optimization problem solved in this case is

$$\underset{\theta}{argmin}\, \ell_{reg}(\theta) = \underset{\theta}{argmin}\, \ell(\theta) + \lambda \mathcal{R}(\theta) \tag{2}$$

While both the methods are effective at reducing overfitting, they present substantial differences, the most relevant being that LASSO introduces sparsity in the solution, meaning that a lot of parameters

are equal to 0 (see Figure 1 for a graphical explanation of this phenomenon) and, hence, they can be removed from the model with the associated predictor. This is desirable in a resource-constrained environment since inference becomes much faster and the model needs less memory to be stored. Minimizing the LASSO-regularized loss presents some challenges due to the non-differentiability of the $L_1$ norm. Instead of applying sub-gradient descent methods, the Alternating Direction Method of Multipliers (ADMM) [25] algorithm has proven to be very effective and is employed in most commercial tools. The idea of ADMM is to express the problem in Equation (2) in the following form:

$$\underset{x}{argmin}\ \ell(\boldsymbol{x}) + \lambda\mathcal{R}(\boldsymbol{z})$$
$$s.t.\ \boldsymbol{x} = \boldsymbol{z} \tag{3}$$

The optimization is then performed in an alternating fashion by minimizing with respect to $\boldsymbol{x}$, $\boldsymbol{z}$, and the dual variables $\boldsymbol{u}$ the following scaled augmented Lagrangian (for more details, see Reference [25]):

$$\mathcal{L}(\boldsymbol{x}, \boldsymbol{z}, \boldsymbol{u}) = \ell(\boldsymbol{x}) + \lambda\mathcal{R}(\boldsymbol{z}) + \frac{\rho}{2}||\boldsymbol{x} - \boldsymbol{z} + \boldsymbol{u}||^2 - \frac{\rho}{2}||\boldsymbol{u}||^2 \tag{4}$$

The iterative algorithm then becomes

$$\boldsymbol{x}^{k+1} = \underset{x}{argmin}\mathcal{L}(\boldsymbol{x}^k, \boldsymbol{z}^k, \boldsymbol{u}^k)$$
$$\boldsymbol{z}^{k+1} = \underset{z}{argmin}\mathcal{L}(\boldsymbol{x}^{k+1}, \boldsymbol{z}^k, \boldsymbol{u}^k)$$
$$\boldsymbol{u}^{k+1} = \boldsymbol{u}^k + \boldsymbol{x}^{k+1} - \boldsymbol{z}^{k+1}$$

An interesting property of this methods is that, using subdifferential calculus [25], we can find a closed form solution for the minimization $\boldsymbol{z}^{k+1} = \underset{z}{argmin}\mathcal{L}(\boldsymbol{x}^k, \boldsymbol{z}^k, \boldsymbol{u}^k)$:

$$\boldsymbol{z}^{k+1} = \mathcal{S}_{\rho/\lambda}\left(\boldsymbol{x}^{k+1} - \boldsymbol{u}^k\right) \tag{5}$$

Here, $\mathcal{S}_\epsilon(\cdot)$ is the soft thresholding operator defined as

$$\mathcal{S}_\epsilon(a) = \begin{cases} a - \epsilon, & \text{if } a > \epsilon \\ 0, & \text{if } |a| \leq \epsilon \\ a + \epsilon, & \text{if } a < -\epsilon \end{cases} \tag{6}$$

It is now clear how sparsity is obtained; in fact, the soft thresholding operator assigns value 0 to elements in absolute value less than the threshold $\epsilon$. Note the soft thresholding is applied element wise on vectors $\boldsymbol{z}^k$.

## 2.2. Multiclass Extensions of Logistic Regression

Logistic regression provides a simple method to perform binary classification; however, in the presence of more than 2 classes, the method cannot be applied. For this reason, various extensions have been proposed that either exploit multiple binary classifiers or define a different function that is able to handle multiple classes. The most common approaches are i) One vs All and ii) softmax regression.

(i) One vs All can be used to extend any binary classifier to a multiclass setting; $K$ different binary classifiers $f_{\theta_k}(\boldsymbol{x})$ are trained on the available data such that the $k$th classifier distinguishes the $k$th class from all the other together. The classification is then performed by assigning every element to the class of which the predictor has the higher value, i.e., $\hat{y}(\boldsymbol{x}) = \underset{k}{argmax}f_{\theta_k}(\boldsymbol{x})$. Unfortunately, this approach loses the probabilistic interpretation that was previously available in the binary set up.

(ii) Softmax regression defines the softmax function $f_k(x) = \frac{e^{\theta_k^T x}}{\sum_{i=1}^{K} e^{\theta_i^T x}}$ to model the multinomial probability distribution of the multiclass problem (note that with $K = 2$ the softmax function is equal to the logistic one). As before, the model is trained by minimizing the cross entropy loss through iterative optimization methods. Remarkably, the probabilistic interpretation here is still valid since $\sum_1^K f_k(x) = 1 \; \forall x$.

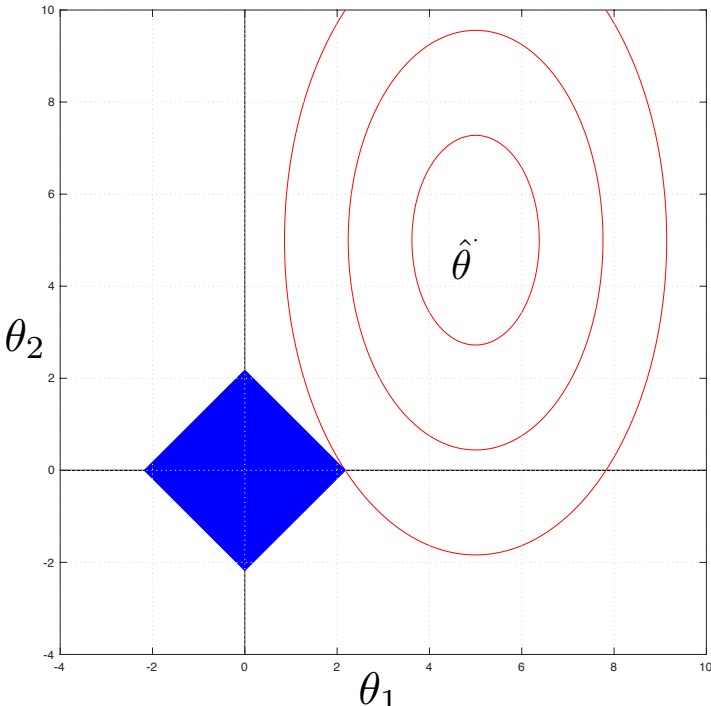

**Figure 1.** Graphical explanation of the sparsity induced by the lasso. In red are the contour lines of the standard loss while in blue is the region where $||\theta||_1 < 2$; during optimization, the optimal point often ends up being on a vertex of the hyper cube.

### 2.3. Multilevel LASSO

It is common in industrial contexts that the processes involved exhibit a hierarchical structure that reflects the behavior of the data and the modeling strategies [26–28]. For example, in our case, the load weight (from 1 kg to 8 kg) is an important discriminator that highly impacts the amount of water absorbed by the laundry and imposes a hierarchical structure to our problem (Figure 2), where we have 8 different leaves associated with different load-weight classes. In order to take into account this structure, one could decide to build a single model for each leaf-node in the hierarchy at the cost of reducing the amount of data available for training. Alternatively, multilevel approaches have been exploited as an intermediate solution where some weights are shared between leaf nodes while others are node specific. Multilevel LASSO [29] belongs to this category. More in detail, a set of features $x^j$ with relative parameters $\theta^j$ are associated to each node of the hierarchy. A path $\mathcal{P}$ is then defined as the set of nodes traversed by an input $x$ in the hierarchy, and the prediction for $x$ is obtained by the following equation:

$$\hat{y}(x) = f\left(\sum_{j \in \mathcal{P}} \theta^{j^T} x^j\right) \tag{7}$$

where $f$ can be any classification function such as logit or softmax. This model can be trained as before by minimizing the cross entropy; moreover, it provides the flexibility to assign different values of the hyperparameter $\lambda$ to different nodes of the hierarchy in order to tune the model thoroughly.

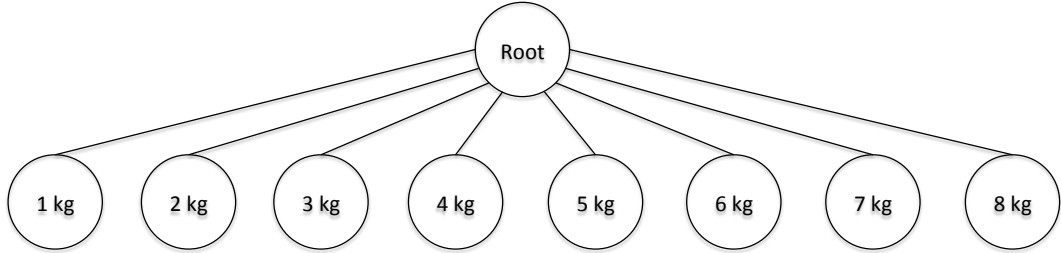

**Figure 2.** Hierarchical structure of our problem.

## 3. Case Study

WMs are extremely common in developed countries, both in domestic and professional environments. It is possible to distinguish two different categories of WMs: vertical axis (Figure 3) and horizontal axis. The first one is characterized by the vertical orientation of the drum rotational axis and is usually popular in the American, Asian, and Australian markets. The second one, instead, is common in the European market and has the drum oriented horizontally. The washing behavior has very distinct characteristics due to the different effects that gravity has on the moving parts of the machines. In particular, when the axis is oriented horizontally, the load moves inside the basket thanks to the combined action of the rotating drum and gravity. The same does not apply in VA-WM, where gravity simply keeps the load at the bottom on the basket and there is the need for an *agitator* that provides the forces necessary for the load to move inside the drum by rapidly rotating clockwise and counterclockwise. The agitator can assume two different shapes (Figure 4): the traditional agitator extends over the entire height of the drum, and the Low Profile Agitator (LPA) instead has a limited height and it requires a smaller amount of water to obtain the same washing performances. For this reason, the producers usually adopt the second technology. Both the agitator and the drum are able to rotate; however, only the agitator is used to wash the clothes while drum movements are used to balance the load and to drain the water at the end of the cycle. Our soft sensor has been developed for a VA-WM equipped with an LPA.

*VA-WM Washing Cycle*

The washing cycle of the VA-WM at hand is composed of three different consecutive phases:

- Warm-Up (WU): During this phase, slow rotations of the drum are performed in order to balance the load. The agitator is locked and does not move.
- Water loading (WL): During this phase, the water is loaded inside the drum.
- Agitation (AG): Here, the actual washing is performed. The drum remains locked, and the agitator rapidly moves the water and, consequently, the clothes, causing the dirt to be removed by friction. The soap is typically added directly inside the drum by the user at the beginning of the cycle.

Our dataset was composed of various laboratory tests performed with different load typologies and weights, where measurements from the Motor Control Board has been acquired during the entire duration of the cycle. Of particular interest is the drum torque and drum speed that reflect the inertial properties of the load and are very informative about its typology. In total, $n = 260$ tests have been performed with four load typologies that, for confidentiality reasons, we will call in an anonymized way types A, B, C, and D. Such tests were performed with laundry weights in $[1\,\text{kg}, 2\,\text{kg}, \ldots, 10\,\text{kg}]$. The size of the available dataset is limited by the time-consuming laboratory tests that require an entire washing cycle to be performed with known weight and typology of the laundry. However, the number of predictors is less than number of samples per class available, which makes our method feasible when combined with regularization methods. In particular, each class has at least 50 tests and the number of predictors in our problem is 50. Hence, the total dataset size is 5 times the number of features used in the model.

In Figure 5, we report an example of time evolution of the drum speed variable. The three washing phases are clearly distinguishable and easily separable by automatic algorithms.

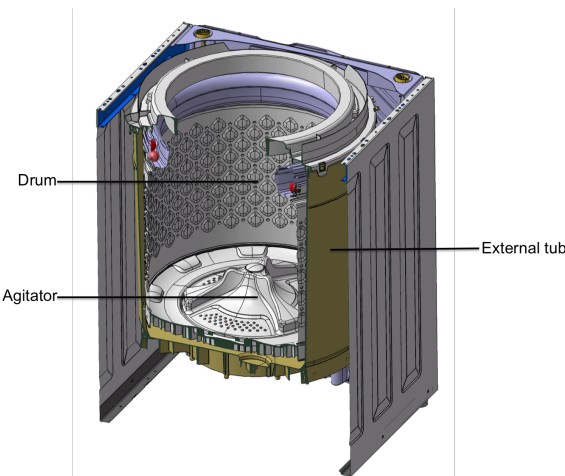

**Figure 3.** Vertical axis washing machine structure: Image provided by Electrolux.

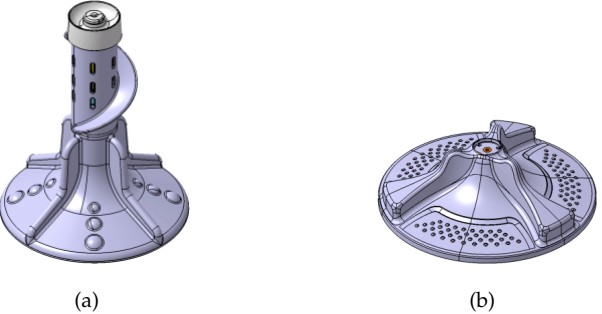

(a)　　　　　　　　　　　　　　　(b)

**Figure 4.** Traditional agitator (**a**) and low profile agitator (**b**): Image provided by Electrolux.

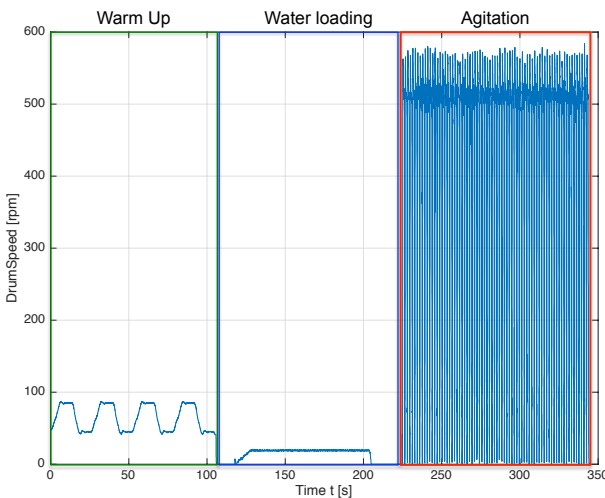

**Figure 5.** Time evolution of the drum speed during a single cycle.

## 4. Load Fabric Detection Method

In this Section, we detail the proposed prediction algorithm with the preliminary feature engineering and extraction phases. As noticeable in Figure 5, the acquired data exhibit a complex

dynamical structure that cannot be directly handled by simple classification algorithms. For this reason, a feature extraction is required where a set of meaningful quantities are extracted from such complex data in order to obtain a set of predictors that can be organized in a design matrix and fed to the ML algorithms. After the feature extraction, we built two prediction models, a simple one (called *Parsimonious Model*) suitable for deployment in the VM firmware and a hierarchical (*Hierarchical Model*) one that is more resource demanding.

### 4.1. Feature Extraction and Analysis

During the feature extraction phase, we determined by visual inspection of the signals some meaningful quantities that allows to discern the different typology classes; typically, these are related to properties of the signals or filtered versions of them such as peaks, transient times, temporal averages, etc. During this process, the water-loading phase has been removed from the data because it depends on the plumbing system and presents substantial variability from home to home and does not provide consistent information.

Due to the extremely different behavior between the WU and the AG phases, we treated them separately. In particular, in the WU phase 4, repeated commutations have analogous dynamics; we thus extracted the same features from each of these commutation. Also, the AG phase is composed of repeated movements with a much faster dynamic called "strokes"; here, we decided to extract some features related to the single strokes and some related to the entire series. In the first case, statistics over the entire series are then computed to obtain a single value for the total phase. At the end of the procedure, a total of 50 features has been extracted from the two phases. It is worth remarking that, only during the AG phase, there is water inside the drum, so we expect this part to be most informative about the water absorption properties of the load and, hence, for our task of typology estimation. In Figure 6, we report two examples of features extracted from the AG phase on the drum torque signal as a function of the weight. It is clearly noticeable the difference between typologies, especially with high weights. However, types A and B show very similar behaviors; this has been justified by domain experts who suggested that the two loads have very similar absorption properties. We thus decided to group together types A and B in a single class called type AB.

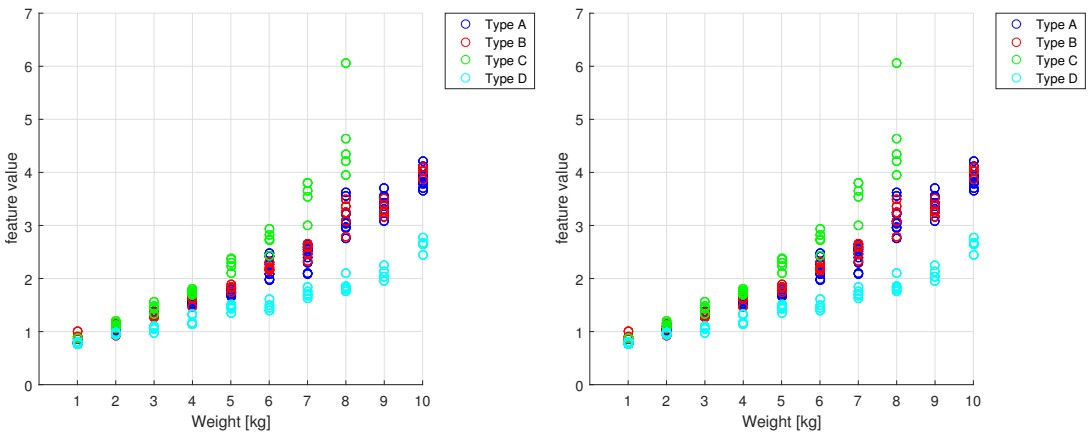

**Figure 6.** Example of features related to the drum torque as a function of the weight.

### 4.2. Modeling

As stated above, the 4 WU commutations provide multiple observations of the same features; hence, it is possible to either employ them on a single model by computing a set of statistics over such features or by building separate models in a bagging fashion [30] and by performing the classification by voting. Preliminary results showed that the former solution presents better performance [18]; hence, we adopted it in all our models.

(i) The *Parsimonious Model* (denoted also as "global" model in the following) was built by employing logistic regression extended with the One vs All method to deal with the multiclass classification problem. The approach has been combined with LASSO regularization, providing a sparse solution feasible for deployment in the firmware.

(ii) The *Hierarchical Model* has been created with the following procedure: since no explicit hierarchical structure was present in the features, we decided to assign to the root node (Figure 2) the 6 features most correlated with the output while the remaining 44 have been assigned to each of the leaf nodes. In this way, the features in the root share the weights with all the leaf nodes and provide most of the information related to the class that can then be refined using the contribution from the remaining predictors. Moreover, the root model should help the classification in the cases where it is more challenging (e.g., low weight). To summarize, the features are organized as follows:

- $p_0 = 6$ features at level 0 (root)
- $p_1 = p_2 = \cdots = p_8 = 44$ features at level 1 (leaves)

Each row of the design matrix $X$ is then defined as follows:

$$x_i = [\bar{x}_i^0, \bar{x}_i^1, \ldots, \bar{x}_i^8] \tag{8}$$

where

$$\bar{x}_i^j = \begin{cases} x_i^j, & \text{if } j \in P_i \\ 0_{1 \times p_j}, & \text{if } j \notin P_i \end{cases} \tag{9}$$

For a graphical explanation of the design matrix creation see Figure 7.

| Features level 0 | Features level 1 | 0 | 0 | 0 | 0 | 0 | 0 | 0 | 1kg |
|---|---|---|---|---|---|---|---|---|---|
| Features level 0 | 0 | Features level 1 | 0 | 0 | 0 | 0 | 0 | 0 | 2kg |
| Features level 0 | 0 | 0 | Features level 1 | 0 | 0 | 0 | 0 | 0 | 3kg |
| ... | ... | ... | ... | ... | ... | ... | ... | ... | ⋮ |
| Features level 0 | 0 | 0 | 0 | 0 | 0 | 0 | 0 | Features level 1 | 8kg |

**Figure 7.** Design matrix for the multilevel LASSO regression model.

Since no data for the type C class was available for 9/10 kg loads, we removed them from the dataset in the multilevel approach.

## 5. Results

We report here the experimental results obtained with the proposed approach. To obtain a statistically significant performance estimation, we employed Monte Carlo Cross Validation (MCCV) [31] with 100 different test/train splits used to test the performance and 100 train/validation splits used for hyperparameter tuning. The results are reported in terms of the classification rate defined as follows: given a test set $\{(x_i, y_i)\ i = 1, \ldots, n_{test}\}$, we compute the predicted class according to the model $\hat{y}_i$ for every $i = 1, \ldots, n_{test}$. Then, let $n_{match}$ be the number of input values where $\hat{y}_i = y_i$; the classification rate (CR) can be computed as follows:

$$CR = \frac{n_{match}}{n_{test}} \cdot 100 \tag{10}$$

### 5.1. Parsimonious Model

In Figure 8, we report the classification rate as a function of the weight of the *Parsimonious Model*. We compare it with simple logistic regression and softmax regression. All three approaches present a CR that is better than chances for every weight; however, the performance at high weights is consistently better, not surprisingly. From the feature analysis, it was noticeable how the different typologies were better distinguishable with heavy loads. Using the estimated weight instead of the real one causes a performance drop, but the CR remains still acceptable. With such similar performances between different methods, the LASSO regularized one is always to be preferred because of its sparsity.

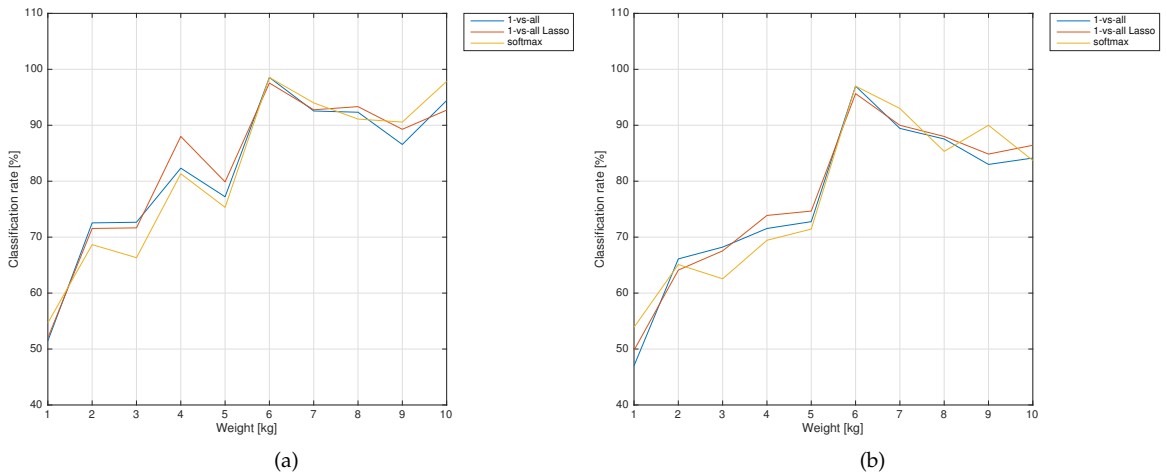

**Figure 8.** Performance of the proposed model with real weight (**a**) and estimated weight (**b**).

### 5.2. Hierarchical Model

In Figure 9, we report the results for the *Hierarchical Model* compared with the *Parsimonious "global" Model* and with a single model for every weight. We notice that, in this case, the *Parsimonious Model* has the best performance overall while the multilevel approach is probably affected by the excessive number of parameters compared to the available data. However, for low weights, it improves the CR of the single-load model, meaning that the shared weights of the root node are helping in this sense. These results highlight the benefit of using a simpler model in cases where a low amount of data is available; however, the multilevel approach can be an interesting alternative for the hierarchical problem, especially where an explicit structure is present also in the features instead of being created artificially, as explained in Section 4.2.

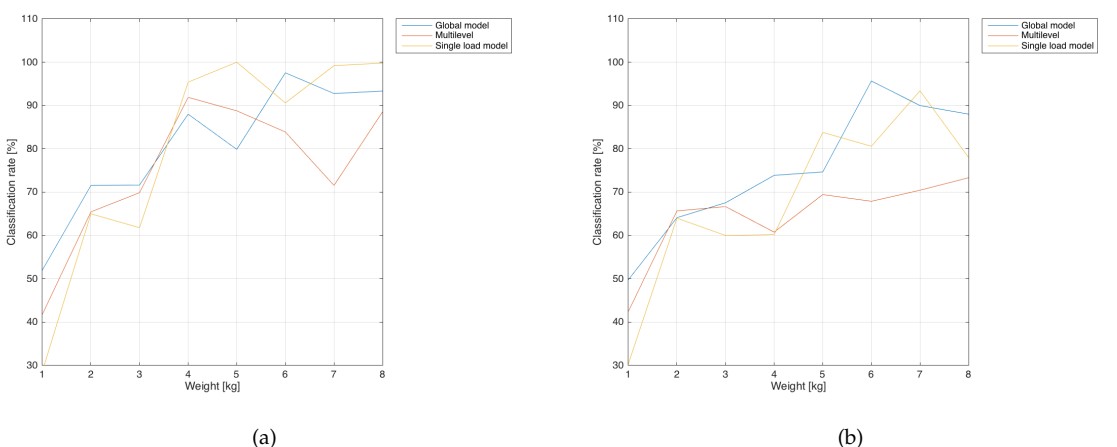

**Figure 9.** Performance of the proposed model with real weight (**a**) and estimated weight (**b**).

## 6. Conclusions

In this paper, we proposed a data-driven soft sensor to detect the laundry typology in VA-WM. The task is very challenging due to the resource-constrained environment, where the model is designed to reside, and to the difficulty in determining the value of an unmeasurable quantity as the load typology. We showed that the drum torque and drum speed variables provide meaningful information about the absorption properties of the load which is directly related to the composition of its fabric. The model achieved interesting performances especially at high weights (>4 kg) where the classification rate was higher than 80% even when the estimated weight was employed. We proposed the use of LASSO regularization, which provides a simple and sparse solution which is easy to implement in the firmware of the WM which has limited computational resources. This decision has been taken with the help of process engineers that are aware of the computational resources available on-board. We showed an alternative hierarchical approach that, however, was not able to achieve comparable performance probably because of the limited amount of data compared to the model size or because of imperfect tuning of the regularization. As shown in the performance comparison, knowing the weight has a considerable impact on the prediction capabilites of our model. The proposed method can be employed to improve user experience by allowing the user to automatically start the laundry process without manually inserting the information about the load and to improve the water/energy consumption by optimizing the cycle depending on the load detected. Both these aspects have huge impacts on the appeal of the product on the market. As a future work, it would be interesting to employ multi-task learning methods where the two estimation models are trained together. For example, an approach based on neural networks would allow common information coming from intermediate layers to be shared between the two tasks in order to improve the quality of the predictions. Moreover, in the home appliances field, there is quite a lot of work in the application of fuzzy control systems [32–35]; hence, an extension of this work is to explore the possibility to employ them in future solutions.

**Author Contributions:** Methodology, M.M. and G.A.S.; validation, E.P. and F.A.; writing—original draft preparation, M.M.; writing—review and editing, E.P., F.A., G.Z., G.A.S.; supervision, E.P., F.A. and G.A.S.; funding acquisition, F.A., A.B. and G.A.S.

**Funding:** Part of this work was supported by MIUR (Italian Minister for Education) under the initiative "Departments of Excellence" (Law 232/2016). This work was also partially funded by Electrolux Italia S.p.A.

**Conflicts of Interest:** The authors declare no conflict of interest. The funders had no role in the design of the study; in the collection, analyses, or interpretation of data; in the writing of the manuscript, or in the decision to publish the results.

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
