# Peer review of "Laundry Fabric Classification in Vertical Axis Washing Machines Using Data-Driven Soft Sensors"

_energies, doi:10.3390/en12214080_

Round 1
Reviewer 1 Report
The literature review is simple, authors should go through the methodologies, assumptions, applicable conditions, or other comparison in literature. Method justification: why did authors suggest logistic regression and LASSO? Since the regularization give a tradeoff between bias and variance of the coefficient of the regressors, authors should clarify this point to justify the regularization. Any details about mathematical formulation of logistic regression and LASSO? Logistic regression and LASSO are linear models, how about nonlinear models? LASSO regularization select important features. The physical meanings of important features should be clarified (eg. causal relationship between regressor and response variable). In case study, authors should show the mean and variance simultaneously about the classification rate based on the MCCV. Based on the variance information, it is possible to justify the results of Figure 7. That is, there is a significant statistical difference among three models. In practice, how to use the proposed model to improve or drive the automation. Authors should give more details about the managerial implications.Author Response
1) The literature review is simple, authors should go through the methodologies, assumptions, applicable conditions, or other comparison in literature.
The literature in soft sensing application for washing machines is very limited and our work is the first that attempts to estimate the load typology from indirect measurements. We have added 2 references related to soft sensor for load humidity estimations in fabric care appliances and some recent works on soft sensing in industrial environments to improve our literature review. (line 65)
2) Method justification: why did authors suggest logistic regression and LASSO? Since the regularization give a tradeoff between bias and variance of the coefficient of the regressors, authors should clarify this point to justify the regularization.
We suggested the use of LASSO because it provides a simple, sparse solution that is easy to implement in the firmware of the WM. This decision has been taken with the help of process engineers that are aware of the computational resources available onboard. We have now stated this clearly in the text (Line 248)
3) Any details about mathematical formulation of logistic regression and LASSO? Logistic regression and LASSO are linear models, how about nonlinear models? LASSO regularization select important features. The physical meanings of important features should be clarified (eg. causal relationship between regressor and response variable).
We have now added a more detailed description of the LASSO, reporting also the algorithm that is typically used to solve such minimization problem (From line 106). As pointed out in the previous response, our algorithm has to be deployed in the firmware of the WM, hence non-linear models add complexity to the method and are not suitable in this application. Unfortunately, we cannot report the important features for confidentiality reasons. However, we have stated in the text that they are derived from the velocity and torque profiles of the motor.
4) In case study, authors should show the mean and variance simultaneously about the classification rate based on the MCCV. Based on the variance information, it is possible to justify the results of Figure 7. That is, there is a significant statistical difference among three models.
We didn't report the variance in the plot because our goal is not to find an absolute best algorithm between the ones proposed but, from Figure 8, we notice that the methods perform comparably, hence the LASSO solution, which is sparse and simple, is the one to be chosen in our resource constrained problem.
5) In practice, how to use the proposed model to improve or drive the automation. Authors should give more details about the managerial implications.
The proposed model can be employed to improve the user experience, by allowing the user to automatically start the laundry process without manually inserting the information about the load, and also the water/energy consumption by optimizing the cycle depending on the load detected. Both these aspects have huge impact on the appeal of the product on the market. We have now added this in the conclusion (line 254).
Reviewer 2 Report
This paper presents a data-driven model to detect the right load typology for washing machine loads. The paper presents a predictive model (logistic regression) to perform the prediction. In addition, they have provided a more complex hierarchy model in order to improve the prediction power.
Overall, the paper sounds very interesting to me as a reader. However, as a researcher active in data-driven models, I was looking to find the features or get a clarification about the target variable. It is still not clear to me what are the dependent and independent variables. If possible, a sample of the data including a description of predictors and the response variable would help much to understand the problem. In addition, one of the main concerns of multiclass classification problems is the lack of available data in each class. Authors mentioned that total there are around 200 samples, but 200 is not strong enough for even binary classification. So, I am eagerly waiting to see if authors have faced any imbalanced classification problem and if so, how they have addressed the problem.
Moreover, when readers deal with manuscripts about home appliances and control problems, usually fuzzy control systems are the first problems to come to mind. But here there is no place to talk about them. It would be great if you have a comparison between data-driven systems and fuzzy systems and explain why you have chosen a data-driven approach.
In total, the problem seems very interesting. However, there are major parts that I believe is missing here. Please address the concerns and I would be happy to read it again.
Regards,
Author Response
1) Overall, the paper sounds very interesting to me as a reader. However, as a researcher active in data-driven models, I was looking to find the features or get a clarification about the target variable. It is still not clear to me what are the dependent and independent variables. If possible, a sample of the data including a description of predictors and the response variable would help much to understand the problem.
We thank the reviewer for the positive comment, unfortunately being the dataset confidential, together with the details related to the feature extraction, we cannot report a description of the predictors. However, we have stated in the text (line 160) that the features are based on the torque and speed of the motor, while the goal is to predict the typology of the load. We were not allowed to report explicitly the different typologies we are trying to distinguish so they were anonymized.
2) In addition, one of the main concerns of multiclass classification problems is the lack of available data in each class. Authors mentioned that total there are around 200 samples, but 200 is not strong enough for even binary classification. So, I am eagerly waiting to see if authors have faced any imbalanced classification problem and if so, how they have addressed the problem.
Dataset of this size are typical environment where data acquisition is expensive, in our case required the execution of 200 lab tests, but the data were well balanced and there are no issues in using simple classification methods such as regularized logistic regression especially since the number of predictors is less than the number of data per class available. We have now clearly stated this in the text (line 164)
3) Moreover, when readers deal with manuscripts about home appliances and control problems, usually fuzzy control systems are the first problems to come to mind. But here there is no place to talk about them. It would be great if you have a comparison between data-driven systems and fuzzy systems and explain why you have chosen a data-driven approach.
We thank the reviewer for the suggestion; however, the goal of this project was not to provide a control algorithm for the machine but to study the feasibility of typology prediction from sensors measurements available on board. We have added in the future works the employment of fuzzy control systems.
Round 2
Reviewer 2 Report
Dear Authors,
Thank you for the revision. The first concern of mine is satisfied with newly added sentences. Regarding the second point about multiclass classification (lines 165-168), please clearly mention that you have enough samples per class and hence, there is no issue with multiclass classification. In addition, it would be good to report the favorite measurement regarding each class. Moreover, "Anyway" (line 167) doesn't belong to the scientific literature! Please use better wording!
Regarding the last point, thanks for adding the comments about fuzzy systems. But it would be great for readers to see some literature regarding the fuzzy control systems. Please add related references to your sentence.
Regards,
Author Response
1) Regarding the second point about multiclass classification (lines 165-168), please clearly mention that you have enough samples per class and hence, there is no issue with multiclass classification. In addition, it would be good to report the favorite measurement regarding each class. Moreover, "Anyway" (line 167) doesn't belong to the scientific literature! Please use better wording!
We have added a statement about the dataset numerosity compared to the number of predictors (line 168). We have also corrected the wording of our sentence.
2) Regarding the last point, thanks for adding the comments about fuzzy systems. But it would be great for readers to see some literature regarding the fuzzy control systems. Please add related references to your sentence.
We have added some references of fuzzy controllers applications in industry and for washing machines.